# Spatial Attention Kinetic Network with E(n)-Equivariance

**Yuanqing Wang**[*] **and John D. Chodera**
Computational and Systems Biology Program,
Sloan Kettering Institute
Memorial Sloan Kettering Cancer Center, New York, N.Y. 10065
yuanqing.wang@choderalab.org

## Abstract

Neural networks that are equivariant to rotations, translations, reflections, and permutations on $n$-dimensional geometric space have shown promise in physical modeling—from modeling potential energy surfaces to forecasting the time evolution of dynamical systems. Current state-of-the-art methods employ spherical harmonics to encode higher-order interactions among particles, which are computationally expensive. In this paper, we propose a simple alternative functional form that uses neurally parametrized linear combinations of edge vectors to achieve equivariance while still universally approximating node environments. Incorporating this insight, we design *spatial attention kinetic networks* with E(n)-equivariance, or SAKE, which are competitive in many-body system modeling tasks while being significantly faster.

## 1 Introduction

Encoding the relevant symmetries of systems of interest into the inductive biases of deep learning architectures has been shown to be crucial in physical modeling. Graph neural networks (GNNs) (Kipf and Welling, 2016; Xu et al., 2018; Gilmer et al., 2017; Hamilton et al., 2017; Battaglia et al., 2018), for instance, preserve permutation equivariance by applying indexing-invariant pooling functions among nodes (particles) and edges (pair-wise interactions) and have emerged to become a powerful workhorse in a wide range of modeling tasks for many-body system (Satorras et al., 2021).

When describing not only the *topology* of the system but also the *geometry* of the state, relevant symmetry groups for three-dimensional systems are SO(3) (rotational equivariance), SE(3) (rotational and translational equivariance), and E(3) (additionally with reflectional equivariance). A ubiquitous and naturally invariant first attempt to encode the geometry of such systems is to employ only radial information, i.e., interparticle distances. This alone has empirically shown utility in predicting quantum chemical potential energies, and considerable effort has been made in the fine-tuning of radial filters to achieve quantum chemical accuracy—1 kcal/mol, the empirical threshold to qualitatively reliably predict the behavior of a quantum mechanical system—and beyond (Schütt et al., 2017).

Nonetheless, radial information alone is not sufficient to fully describe node environments—the spatial distribution of neighbors around individual particles. The relative locations of particles around a central node could drastically change despite maintaining distances to these neighbors unaltered. To describe node environments with completeness, one needs to address these remaining degrees of freedom. Current state-of-the-art approaches encode angular distributions by employing a truncated series of spherical harmonics to generate higher-order feature representations; while these models have been shown to be data efficient for learning properties of physical systems, these features are expensive to compute, with the expense growing rapidly with the order of harmonics included (Thomas et al., 2018; Klicpera et al., 2021a; Fuchs et al., 2020; Batzner et al., 2021; Anderson et al., 2019). The prohibitive cost would prevent this otherwise performant class of models

---

[*]Alternative address: Ph.D. Program in Physiology, Biophysics, and System Biology, Weill Cornell Medical College, Cornell Univerisity, New York, N.Y. 10065

from being employed in materials and drug design, where rapid simulations of large systems are crucial to provide quantitative insights.

Here, we design a simple functional form, which we call *spatial attention*, that uses the norm of a set of neurally parametrized linear combinations of edge vectors to describe the node environment. Though simple in form, easy to engineer, and ultra-fast to compute, spatial attention is capable of universally approximating any functions defined on local node environment while preserving E(n)-invariance/equivariance in arbitrary $n$-dimensional space.

After demonstrating the approximation universality and invariance of spatial attention, we incorporate it into a novel neural network architecture that uses spatial attention to parametrize fictitious velocity and positions equivariantly, which we call a *spatial attention kinetic network with E(n)-Equivariance*, or SAKE (pronounced *saké* (*sah-keh*), like the Japanese rice wine)[1]. To demonstrate the robustness and versatility of SAKE, we benchmark its performance on potential energy approximation and dynamical system forecasting and sampling tasks. For all popular benchmarks, compared to state-of-the-art models, SAKE achieves competitive performance on a wide range of invariant (MD17: Table 1, QM9: Table 3, ISO17: Table 2) and equivariant (N-body charged particle Table 4, walking motion: Table 6) while requiring only a fraction of their training and inference time.

## 2    BACKGROUND

In this section, we provide some theoretical background on physical modeling, equivariance, and graph neural networks to lay the groundwork for the exposition of spatial attention networks.

### 2.1    EQUIVARIANCE: PERMUTATIONAL, ROTATIONAL, TRANSLATIONAL, AND REFLECTIONAL

A function $f : \mathcal{X} \to \mathcal{Y}$ is said to be equivariant to a symmetry group $G$ if

$$f(T_g(\mathbf{x})) = S_g(f(\mathbf{x})) \tag{1}$$

$\forall g \in G$ and some equivalent transformations on the two spaces respectively $T_g : \mathcal{X} \to \mathcal{X}$ and $S_g : \mathcal{Y} \to \mathcal{Y}$.

If on a $n$-dimensional space $\mathcal{X} = \mathcal{Y} = \mathbb{R}^n$, we have permutation $P$ and $T_g(\mathbf{x}) = P\mathbf{x}, S_g(\mathbf{y}) = P\mathbf{y}$ satisfying Equation 1, we say $f$ is permutationally equivariant; if $T_g(\mathbf{x}) = \mathbf{x}R$ where $R \in \mathbb{R}^{n \times n}$ is a rotation matrix $RR^T = I$, and $S_g(\mathbf{y}) = \mathbf{y}R$ we say $f$ is rotationally equivariant; if $T_g(\mathbf{x}) = \mathbf{x} + \Delta\mathbf{x}$ and $S_g(\mathbf{y}) = \mathbf{y} + \Delta\mathbf{x}$, where $\mathbf{x} \in \mathbb{R}^n$ we say $f$ is translationally equivariant; finally, if $T_g(\mathbf{x}) = \text{Ref}_\theta(\mathbf{x})$ and $S_g(\mathbf{y}) = \text{Ref}_\theta(\mathbf{y})$, and $\text{Ref}_\theta$ is a reflection on $n$-dimensional space, we say $f$ is reflectionally equivariant.

### 2.2    GRAPH NEURAL NETWORKS

Modern GNNs, which exchanges and summarizes information among nodes and edges, are better analyzed through the *spatial* rather than *spectral* lens, according to Wu et al. (2019)'s classification. Following the framework from Gilmer et al. (2017); Xu et al. (2018); Battaglia et al. (2018), for a node $v$ with neighbors $u \in \mathcal{N}(v)$, in a graph $\mathcal{G}$, with $h_v^{(k)}$ denoting the feature of node $v$ at the $k$-th layer (or $k$-th round of message-passing) and $h_v^0 \in \mathbb{R}^C$ the initial node feature on the embedding space, the $k$-th message-passing step of a GNN can be written as three steps:

First, an *edge update*,

$$h_{e_{uv}}^{(k+1)} = \phi^e\big(h_u^{(k)}, h_v^{(k)}, h_{e_{uv}}^{(k)}\big), \tag{2}$$

where the feature embeddings $h_u$ of two connected nodes $u$ and $v$ update their edge feature embedding $h_{e_{uv}}$, followed by *neighborhood aggregation*,

$$a_v^{(k+1)} = \rho^{e \to v}(\{h_{e_{uv}}^{(k)}, u \in \mathcal{N}(v)\}), \tag{3}$$

where edges incident to a node $v$ pool their embeddings to form *aggregated neighbor embedding $a_v$*, and finally a *node update*,

$$h_v^{(k+1)} = \phi^v(a_v^{(k+1)}, h_v^{(k)}) \tag{4}$$

---

[1]Implementation: https://github.com/choderalab/sake

where $\mathcal{N}(\cdot)$ denotes the operation to return the multiset of neighbors of a node and $\phi^e$ and $\phi^v$ are implemented as feedforward neural networks. Since the neighborhood aggregation functions $\rho^{e \rightarrow v}$ are always chosen to be indexing-invariant functions, namely a SUM or a MEAN operator, Equation 3, and thereby the entire scheme, is permutationally invariant.

**Problem Statement.** We are interested in designing a class of parametrized functions $f_\theta : \mathcal{X} \times \mathcal{H} \rightarrow \mathcal{X} \times \mathcal{H}$ that map from and to the joint spaces of ($n$-dimensional) geometry $\mathcal{X} \in \mathbb{R}^n$ and semantic embedding $\mathcal{H} \in \mathbb{R}^C$ such that it is permutationally, rotationally, translationally, and reflectionally equivariant on $\mathcal{X}$ and invariant on $\mathcal{H}$. That is, for a given coördinate $\mathbf{x} \in \mathcal{X}$, embedding $h \in \mathcal{H}$ and any transformation mentioned in Section 2.1 $T : \mathcal{X} \rightarrow \mathcal{X}$ (rotation, translation, and reflection), we have:

$$\mathbf{x}_f, h_f = f_\theta(\mathbf{x}, h) \Longleftrightarrow T(\mathbf{x}_f), h_f = f_\theta(T(\mathbf{x}), h) \tag{5}$$

# 3 RELATED WORK: INVARIANT AND EQUIVARIANT GNNs

**Invariant GNN.** Although the notion of graph might not appear in their original publications, a plethora of neural architectures could be viewed as *de facto* graph neural networks with invariance on geometric space (for a survey, see (Han et al., 2022)). SchNet (Schütt et al., 2017), for example, approximates the potential energy of molecules by applying radially-symmetric convolutions that only operate on the distances between atoms to ensure the model is E(3)-invariant, effectively using a graph neural network architecture where nodes denote atoms and edges denote the distance separation between pairs of atoms within a cutoff radius. In addition to interatomic distances, invariant models could also incorporate information about the angular distribution of atom neighbors by computing atomic features that incorporate angular information of neighbor-atom-neighbor triplets (Smith et al., 2017; Behler, 2011; Klicpera et al., 2020a; Wang et al., 2022; 2023; Liu et al., 2022). Compared to SAKE, these models are not capable of *equivariant* modeling; even in an invariant setting, they are empirically less performant (See invariant tasks performance Section 6.1).

**Equivariant GNN with spherical harmonics.** Architectures achieving SE(3)-*equivariance* by leveraging Bessel functions and spherical harmonics to encode higher-order interactions (Thomas et al., 2018; Klicpera et al., 2021a; Fuchs et al., 2020; Anderson et al., 2019; Klicpera et al., 2021b; Brandstetter et al., 2021; Liu et al., 2022) shows outstanding performance and data efficiency, some of which is on par with SAKE (Table 1, 4). Villar et al. (2021) discusses that these higher-order representation is not necessary to construct invariantly and equivariantly universal models. Meanwhile, they tend to be difficult to engineer and expensive to train and run. (See runtime benchmarks in aforementioned tables.)

**Equivariant GNN with dot product scalarization.** Recent efforts (Schütt et al., 2021; Thölke and Fabritiis, 2022; Huang et al., 2022) relying on dot product of edge tensors as equivariant nonlinearity have achieved competitive results on machine learning potential construction, among which none has been validated to perform well on both *equivariant* and *invariant tasks*. Experimentally, they are consistently outperformed by SAKE in invariant (Section 6.1: Table 1) and equivariant (Section 6.2: Table 6) tasks.

**Message passing between emebedding and geometric spaces.** Satorras et al. (2021) has formalized the link between graph neural networks and and equivariance and provided a generalizing framework encompassing iterative geometric-to-embedding and embedding-to-geometric type update. Like our proposed architecture, E(N) Equivariant Graph Neural Networks (EGNN), proposed in Satorras et al. (2021), also uniquely describes the geometry of a $n$-body system, although their argument was based on a *global* scale given sufficient steps of message passing, while we can sufficiently describe the *local* geometric environment. This advantage is evidenced by extensive experiments (invariant: Table 3; equivariant: Table 4 and 6). In ablation study (Section 6.3), we show that without *spatial attention*, EGNN is not competitive in potential energy modeling, even when all other tricks used in this paper were added. Similar to EGNN (Satorras et al., 2021), our model is equivariant w.r.t. a general E(n) group and are not restricted to E(3). Also worth noting is that the Geometric Vector Perceptrons algorithm proposed in Jing et al. (2021) could be regarded as a special case of our framework where the attention weights are learned globally, whereas we learn them in an amortized manner and thus can be transductively generalized across systems.

## 4 THEORY: SPATIAL ATTENTION

Given a node $v$ with embedding $h_v \in \mathcal{H} = \mathbb{R}^C$ (where $C$ denotes the embedding dimension) and position $\mathbf{x}_v \in \mathcal{X} = \mathbb{R}^n$ (where $n$ denotes the geometry dimension) in a graph $\mathcal{G}$, its neighbors $u \in \mathcal{N}(v)$ with connecting edges $\{e_{uv}\}$, with displacement vector $\vec{e}_{uv} = \mathbf{x}_v - \mathbf{x}_u$ and embedding $h_{e_{uv}} = \rho^{v \to e}(h_v, h_u)$ with some aggregation function $\rho^{v \to e}$, we define spatial attention $\phi : \mathcal{X} \times \mathcal{H} \to \mathcal{H}$ as

$$\phi^{\text{SA}}(v) = \mu\big(\bigoplus_{i=1}^{N_\lambda} || \sum_{u \in \mathcal{N}(v)} \lambda_i(h_{e_{uv}}) f(\vec{e}_{uv})||\big), \tag{6}$$

where $\lambda_i : \mathcal{H} \to \mathbb{R}^1, i = 1, ..., N_\lambda$ is a set of arbitrary attention weights-generating function that operates on the edge embedding, $f : \mathcal{X} \to \mathcal{X}$ is an *equivariant* function that operate on the edge vector, $\mu : N_\lambda \to \mathcal{H}$ is an arbitrary function that takes the norms of $N_\lambda$ linear combinations, and $\oplus$ denotes concatenation. We drop the explicit dependence of $\phi^{\text{SA}}(v)$ on both geometric and embedding properties of $v$ and $u$ for simplicity. We name this functional form *attention* because it characterizes the alignment between edge vectors in various projections.

When implemented, $\lambda$ and $\mu$ take the form of feed-forward neural networks. In other words, for each node $v$, we first parametrize a $N_\lambda = |\{\lambda_i\}|$ (is analogous to the number *heads* in multi-head attention) sets of attention weights based on the edge embeddings $\lambda_i(h_{e_{uv}})$. At the same time, we also equivariantly (on E(n) group) transform the edge vector into $f(\vec{e}_{uv})$. Next, we use those $N_\lambda$ sets of attention weights to linearly combine these vectors among edges and take the Euclidean norm of each of these linear combinations, resulting in $N_\lambda$ norms. Lastly, we concatenate these norms and put them into a feed-forward neural network $\mu$ to compute the node embedding. It naturally follows that:

**Remark 1.** *Spatial attention is permutationally, rotationally, transalationally, and reflectionally invariant on E(n),*

since $h_{e_{uv}}$ is invariant w.r.t. indexing and the Euclidean norm function is invariant to rotation, translation, and reflection.

With all local degrees of freedom incorporated in spatial attention, it is perhaps intuitive to see that $\phi^{\text{SA}}(v)$ can uniquely define the local environment of nodes up to E(n) symmetry. We formalize this finding (Proof in Appendix Section 8.1):

**Theorem 1.** *For a node $v$ in a graph $\mathcal{G}$ with neighbors $u \in \mathcal{N}(v)$ connected to $u$ by edges with positions $\mathbf{x}_v$ and $\mathbf{x_u}$ distinct embeddings $h_{e_{vu_i}} \neq h_{e_{vu_j}}, \forall 1 \leq i, j \leq |\mathcal{N}(v)|$, and any E(n)-invariant continuous function $g(\mathbf{x}_v, \mathbf{x}_{u_i}|h_{e_{vu_i}})$ spatial attention $\phi^{\text{SA}}$ can approximate g with arbitrarily small error $\epsilon$ with some $\{\lambda\}$, f, and $\mu$.*

It is worth noting here that we only consider the universal approximation power of a mapping from the geometry space $\mathcal{X}$ to a scalar space, without considering that on the space of node (and edge) embedding; regardless of geometry, GNNs that operate on neighborhood node embeddings are generally not universal (Xu et al., 2018; Corso et al., 2020). This universal approximative power on $E(n)$-invariant functions can lead to universally approximative parametrization of $E(n)$-equivariant functions, which we show in Remark 2. Practically, the inequality condition is not difficult to satisfy: we implement $h_e$ as dependent upon edge length so even if the semantic embedding of edges are identical, the inequality $h_{e_{vu_i}} \neq h_{e_{vu_j}}$ holds as long as the system is not strictly symemtrical at all times (namely the mirror symmetry presented in hydrogen molecule) despite distortions resulted from vibrations.

## 5 ARCHITECTURE: SPATIAL ATTENTION KINETIC NETWORK (SAKE)

Leveraging the simple functional form introduced in Section 4, we design a fast, efficient, and easy-to-implement architecture termed a *spatial attention kinetic network with E(n)-equivariance*, or SAKE. Here, we introduce the remaining components of SAKE before assembling them in a modular way. The necessity of these designs are experimentally justified with ablation study in Section 6.3.

**Edge embedding.** To embed pairwise interactions, we combine the SchNet (Schütt et al., 2017)-style continuous filter convolution and the simple concatenation of the scalar-valued distance as in

---

**Algorithm 1** Spatial Attention Kinetic Networks Layer

---

    **function** SAKELAYER($\{h_v^{(k)}\}, \{\mathbf{x}_v^{(k)}\}, \{\mathbf{v}_v^{(k)}\}, \mathcal{G}$)              ▷ Input embedding, position, and velocity
        **for** $v \in \mathcal{V}$ **do**
            **for** $u \in \mathcal{N}(v)$ **do**
                $h_{e_{uv}}^{(k)} \leftarrow \phi^e(h_v^k, h_u^k, ||\vec{\mathbf{e}}_{uv}||)$               ▷ Edge update, Sec 5 Eq 7
            **end for**
            $h_{e_{uv}}^{(k+1)} \leftarrow h_{e_{uv}}^{(k)} * \alpha_{uv}^{\mathcal{X} \times \mathcal{H}}$        ▷ Semantic attention and distance cutoff, Sec 5 Eq 10
            **for** $u \in \mathcal{N}(v)$ **do**
                $h_{SA_v}^{(k+1)} = \phi^{SA}(h_{e_{uv}}^{(k)}, \vec{\mathbf{e}}_{uv})$             ▷ Spatial attention, Sec 5 Eq 6
            **end for**
            $a_v^{(k)} \leftarrow \sum\limits_{u \in \mathcal{N}(v)} h_{e_{uv}}^{(k)}$           ▷ Neighborhood aggregation, Sec 2.2 Eq 3
            $\mathbf{v}_v^{(k+1)} \leftarrow \phi^{v \to \mathcal{V}}(h_v^{(k)})\mathbf{v}^{(k)} + \mathbf{W}_v \sum\limits_i \sum\limits_{u \in \mathcal{N}(v)} \lambda_i(h_{e_{uv}}^{(k)})f(\vec{\mathbf{e}}_{uv}^k)$    ▷ Vel. update, Sec 5 Eq 12
            $\mathbf{x}_v^{(K)} \leftarrow \mathbf{x}_v^{(k)} + \mathbf{v}_v^{(K)}$               ▷ Position update, Sec 5 Eq 12
            $h_v^{(k+1)} \leftarrow \phi^v(h_v^{(k)}, a_v^{(k)}, h_{SA_v}^{(k)})$            ▷ Node update, Sec 2.2 Eq 4
            **return** $\{h_v^{(k+1)}\}, \{\mathbf{x}_v^{(k+1)}\}, \{\mathbf{v}_v^{(k+1)}\}$
        **end for**
    **end function**

---

Satorras et al. (2021) to achieve the balance between high radial resolution and large receptive field. The resulting edge embedding is thus

$$h_{e_{uv}}^{(k)} = \phi^e(h_u^{(k)} \oplus h_v^{(k)} \oplus ||\vec{\mathbf{e}}_{uv}^{(k)}|| \oplus \text{RBF}(||\vec{\mathbf{e}}_{uv}^{(k)}||) \odot f^r(h_u^{(k)} \oplus h_v^{(k)})), \tag{7}$$

where $\odot$ denotes Hadamard product and $f^r$ is a (filter-generating) feed-forward network as in Schütt et al. (2017).

**Semantic attention and distance cutoff.**    To promote anisotropy (which Dwivedi et al. (2020) finds useful in GNNs) in the pooling operation, apart from spatial attention introduced in Section 4, we compute the attention score on semantic and geometry space to weight interactions among particles based on embedding similarity and distances on $n$-dimensional space. The distance weights are calculated using the cutoff function proposed in Unke and Meuwly (2019):

$$\alpha_{u_i v}^{\mathcal{X}} = \begin{cases} \frac{1}{2}\cos(\frac{\pi||\vec{\mathbf{e}}_{u_i v}||}{d_0} + 1), ||\vec{\mathbf{e}}_{u_i v}|| \leq d_0; \\ 0, ||\vec{\mathbf{e}}_{u_i v}|| > d_0, \end{cases} \tag{8}$$

to filter out interactions outside a certain range ($d_0$). And semantic attention weights $\alpha_{u_i v}^{\mathcal{H}}$ are calculated similar to Graph Attention Networks (Veličković et al., 2018),

$$\alpha_{u_i v}^{\mathcal{H}} = \frac{\exp(\sigma(\mathbf{a}^T h_{e_{u_i v}}))}{\sum \exp(\sigma(\mathbf{a}^T h_{e_{uv}}))}. \tag{9}$$

To produce models for the purpose of molecular simulation by ensuring continuous forces and gradients, one would need to choose as sigma $\sigma$ as at least C2-continuous activation functions, namely CeLU (Barron, 2017). These weights are combined and normalzied:

$$\alpha_{u_i v}^{\mathcal{X} \times \mathcal{H}} = \frac{\alpha_{u_i v}^{\mathcal{X}} \alpha_{u_i v}^{\mathcal{H}}}{\sum \alpha_{uv}^{\mathcal{X}} \alpha_{uv}^{\mathcal{H}}}. \tag{10}$$

It is trivial to expand to multi-head attention with $k$ sets of $(\mathbf{a}_{1,\dots,k}^T)$ and the resulting combined attention weights concatenated.

Since the edge embedding after mixed Euclidean and semantic attention $h_{e_{uv}} * \alpha_{uv}^{\mathcal{X} \times \mathcal{H}}$ already encodes the desired inductive bias that nodes farther away from each other would have less impact on each other's embedding, we directly use this representation and simply set $\lambda_i$ as a linear projection (with weights $\mathbf{W}_\lambda$) and $f$ as identity in Equation 6:

$$\lambda_i(h_{e_{uv}}) = \mathbf{W}_\lambda h_{e_{uv}}; f(\vec{\mathbf{e}}_{uv}) = \vec{\mathbf{e}}_{uv}/||\vec{\mathbf{e}}_{uv}||. \tag{11}$$

We leave more elaborate choices of $\lambda$ and $f$ functions for future study.

**Fictitious velocity integration.** Similar to Satorras et al. (2021), we keep track of a fictitious velocity variable $\mathbf{v}_v$ for each node and linearly combine it (with weights $\mathbf{W}_\lambda$) update it and positions $\mathbf{x}_v$ in turns *like* a Euler-discretized Hamiltonian integration.

$$\mathbf{v}_v^{(k+1)} = \phi^{v \to \mathcal{V}}(h_v^{(k)})\mathbf{v}^{(k)} + \mathbf{W}_v \sum_i \sum_{u \in \mathcal{N}(v)} \lambda_i(h_{e_{uv}}^{(k)})f(\vec{\mathbf{e}}_{uv}^k) \tag{12}$$

$$\mathbf{x}_v^{(k+1)} = \mathbf{x}_v^{(k)} + \mathbf{v}_v^{(k)} \tag{13}$$

During velocity update, we scale the current velocity and, for the sake of parameter saving, reuse the same set of linear combinations of edge vectors used in Equation 6 as the additive term.

**Remark 2.** *If $h_{e_{uv}}$ is distinct among edges, there exist sets of $\lambda_i$ (even when $f = I$ is the identity) such that Equation 12 can approximate any vector on the subspace spanned by edge vectors and is thus universal up to equivariance thereon.*

**Equivariance analysis.** We inlay these modular components into the framework of graph neural networks (described in Section 2.2) to produce the SAKE architecture, which we show in Algorithm 1. (A SAKEModel is defined by applying SAKELayer iteratively from $k = 0$ to $k = K - 1$, the depth of the network.) We have previously discussed that spatial attention is E(n)-invariant in Remark 4. With the E(n) equivariance of velocity update and position update (Equation 12) being proved in Satorras et al. (2021) and the rest of the model only takes the norm of edges and is E(n) invariant, SAKE is E(n)-equivariant.

**Runtime analysis.** The runtime complexity of spatial attention (Equation 6), which is the bottleneck of this algorithm, is $\mathcal{O}(|\mathcal{E}|N_\lambda CD)$, i.e., linear in the number of graph edges $|\mathcal{E}|$, number of attention weights $N_\lambda$, embedding dimension $C$, and geometric dimension $D$. When implemented, the FOR loops in Algorithm 1 can be packed into multi-dimensional tensors that could benefit from GPU acceleration.

**Relation to spherical harmonics-based models.** Under the framework of TFN (Thomas et al., 2018), the embedding and position input in Algorithm 1 $\{h_v^{(k)}\}$, $\{\mathbf{x}_v^{(k)}\}$ corresponds to the $l = 0$ and $l = 1$ type tensors. The concatenation followed by neural network in Equation 6 is loosely analogous to the direct sum operation in Clebsch-Gordon decomposition. While Smith et al. (2017); Schütt et al. (2017); Satorras et al. (2021) operates on $l = 0$ tensors only, the *spatial attention* mechanism we propose (Section 4 Equation 6) and velocity/position update (Section 5 Equation 12) corresponds to $1 \oplus 1 \to 0$ and $1 \oplus 0 \to 1$ type networks, respectively. Klicpera et al. (2021a); Villar et al. (2021) have discovered that $l = 0, 1$ are the complete levels of tensor to universally describe geometry on $E(3)$, while higher-order tensors are not necessary to completely describe the node environment. Kovács et al. (2021) also discusses the concept of *density projection* where body-order functional forms can be recovered by lower-order terms.

## 6 EXPERIMENTS

As discussed in Section 2, SAKE provides a mapping from and to the joint space of geometry and embedding $\mathcal{X} \times \mathcal{H}$, while being equivariant on geometric space and invariant on embedding space. We are therefore interested to characterize the performance of SAKE on two types of tasks: *invariant modeling* (Section 6.1), where we model some scalar property of a physical system, namely potential energy; and *equivariant modeling* (Section 6.2), where we predict coördinates conditioned on initial position, velocity, and embedding.

On both classes of tasks, SAKE displays competitive performance while requiring significantly less inference time compared to current state-of-the-art models. See Appendix Section 9 for experimental details and settings.

### 6.1 INVARIANT TASKS: MACHINE LEARNING POTENTIAL CONSTRUCTION

**MD17 potential energy** (Chmiela et al., 2017) tests the capacity of the model in the extreme small data regime. Its training set contains merely 1000 configurations and quantum chemical energies and forces of single small molecules in vacuum computed using density functional theory (DFT). As

Table 1: Inference time (ms) and test set energy (E) and force (F) mean absolute error (MAE) (meV and meV/Å) on the MD17 quantum chemical dataset.

| | | SchNet
Schütt et al., 2017 | DimeNet
Klicpera et al., 2020b | sGDML
Chmiela et al., 2019 | PaiNN
Schütt et al., 2021 | GemNet(T/Q)
Klicpera et al., 2021a | NequIP
Batzner et al., 2021 | SAKE |
|---|---|---|---|---|---|---|---|---|
| Inference time | batch of 32 | | 65 | | | 88/376 | 206 | **12** |
| | batch of 4 | | 31 | | | 38/99 | 197 | **4** |
| Aspirin | E | 16.0 | 8.8 | 8.2 | 6.9 | - | **5.3** | $5.91^{5.92}_{5.88}$ |
| | F | 58.5 | 21.6 | 29.5 | 14.7 | 9.4 | **8.2** | $8.09^{8.10}_{8.08}$ |
| Ethanol | E | 3.5 | 2.8 | 3.0 | 2.7 | - | **2.2** | $\mathbf{2.20}^{2.20}_{2.20}$ |
| | F | 16.9 | 10.0 | 14.3 | 9.7 | 3.7 | 3.8 | $\mathbf{2.75}^{2.75}_{2.75}$ |
| Malonaldehyde | E | 5.6 | 4.5 | 4.3 | 3.9 | - | **3.3** | $3.22^{3.23}_{3.21}$ |
| | F | 28.6 | 16.6 | 17.8 | 13.8 | 6.7 | 5.8 | $\mathbf{4.32}^{4.32}_{4.31}$ |
| Naphtalene | E | 6.9 | 5.3 | 5.2 | 5.0 | - | **4.9** | $4.91^{4.92}_{4.91}$ |
| | F | 25.2 | 9.3 | 4.8 | 3.3 | 2.2 | 1.6 | $2.25^{2.25}_{2.25}$ |
| Salicylic acid | E | 8.7 | 5.8 | 5.2 | 4.9 | - | **4.0** | $4.67^{4.67}_{4.66}$ |
| | F | 36.9 | 16.2 | 12.1 | 8.5 | 5.4 | 3.9 | $4.29^{4.30}_{4.28}$ |
| Toluene | E | 5.2 | 4.4 | 4.3 | 4.1 | - | **4.0** | $4.00^{4.01}_{4.00}$ |
| | F | 24.7 | 9.4 | 6.1 | 4.1 | 2.6 | 2.0 | $2.10^{2.10}_{2.10}$ |
| Uracil | E | 6.1 | 5.0 | 4.8 | 4.5 | - | **4.5** | $4.51^{4.52}_{4.50}$ |
| | F | 24.3 | 13.1 | 10.4 | 6.0 | 4.2 | 3.3 | $4.26^{4.28}_{4.25}$ |

summarized in Table 1, SAKE consistently outperforms all benchmarked models with the exception of NequIP (Batzner et al., 2021), which has comparable performance but is noticeably slower. (Note that, to avoid inaccurate report on the suboptimal software configuration, we only report runtime data directly quoted from their original publications and replicate the hardware environment by ourselves. The same applies hereafter.) In terms of training cost, most state-of-the-art models requires days of training, whereas the MD17 experiment was completed within 6 hours. This significant advantage in speed would allow SAKE to be more rapidly trained and deployed on realistic applications involving molecular dynamics simulations.

**ISO17** (Schütt et al., 2017) goes beyond the single-molecule regime. It involves a slightly more diverse chemical space containing 5000-step *ab initio* molecular dynamics simulation trajectories (with energies and forces) of 129 molecules with the same formula $C_7H_{10}O_2$. The test set of ISO17 is split into *known* (which Kovács et al. (2021) argues to be very close to training set) and *unknown* molecules, based on the chemical identity (topology) at the beginning of the simulation, which could be regarded as interpolative and extrapolative tasks, respectively. As shown in Table 2, SAKE significantly outperforms other models on the unknown molecules in the test set, indicating that SAKE is capable of extrapolating and generalizing onto unseen chemical spaces when trained on limited data.

**QM9** (Ramakrishnan et al., 2014) tests the transductive generalizability across distinct small chemical graphs. It entails a very diverse chemical space of 134k molecules with annotated phsyical properties calculated with B3LYP/6-31G(2df,p) level of quantum chemistry, albeit all with at-equilibrium (low energy) conformations. In Table 3, SAKE achieves state-of-the-art performance at predicting HOMO, LUMO, and $\Delta\epsilon$ properties—a class of most crucial molecular properties closely related to reactivity. Interestingly, SAKE performs competitively on predicting *extensive* physical properties but not *intensive* ones (also see discussions in Pronobis et al. (2018)). We hypothesize that this will be mitigated by choosing size-invariant pooling functions—we leave this for future study.

## 6.2 EQUIVARIANT TASKS

**Charged N-body dynamical system forecasting** (Kipf et al., 2018; Fuchs et al., 2020) tests if a model can predict the evolution of a physical system sufficiently long after initial conditions. This simple system consists of 5 charge-carrying particles with initial positions and velocities, and the position at a given moment is predicted. As shown in Table 4, although the interactions (Coulomb forces) are entirely pairwise, we see here that the additional expressiveness SAKE affords lead to competitive performance on this demonstrative task.

Table 2: Test set energy (E) and force (F) mean absolute error (MAE) (meV and meV/Å) on known and unknown molecules in ISO17.

| | | ACE (Kovács et al., 2021) | SchNet (Schütt et al., 2017) | PhysNet (Unke and Meuwly, 2019) | SAKE |
|---|---|---|---|---|---|
| *known* | E | 16 | 16 | **4** | $12.17_{12.12}^{12.18}$ |
| | F | 43 | 43 | **5** | $12.33_{12.31}^{12.34}$ |
| *unknown* | E | 85 | 104 | 127 | $\mathbf{53.37}_{53.15}^{53.62}$ |
| | F | 85 | 95 | 60 | $\mathbf{39.46}_{39.35}^{39.59}$ |

Table 3: QM9 test set performance (mean absolute error).

| | $\alpha$ Bohr$^3$ | $\Delta\epsilon$ meV | HOMO meV | LUMO meV | $\mu$ D | $C_v$ cal/mol K |
|---|---|---|---|---|---|---|
| SchNet (Schütt et al., 2017) | 0.235 | 63 | 41 | 34 | 0.033 | 0.033 |
| DimeNet++ (Klicpera et al., 2020a) | 0.044 | 33 | 25 | 20 | 0.030 | 0.023 |
| SE(3)-TF (Fuchs et al., 2020) | 0.142 | 53 | 35 | 33 | 0.051 | 0.054 |
| EGNN (Satorras et al., 2021) | 0.071 | 48 | 29 | 25 | 0.029 | 0.031 |
| PaiNN (Schütt et al., 2021) | 0.059 | 36 | 46 | 20 | **0.012** | **0.024** |
| TorchMD-Net (Thölke and Fabritiis, 2022) | 0.059 | 36 | 20 | 17 | **0.011** | **0.023** |
| SphereNet (Liu et al., 2022) | **0.030** | 31 | 19 | 23 | 0.025 | **0.022** |
| SAKE | 0.068 | **23** | **16** | **13** | **0.014** | 0.087 |

**MD17 forecast**   (Chmiela et al., 2017; Huang et al., 2022) involves a simulation forecast task with slightly more complicated systems compared to Table 4. It uses the same dataset in Table 1, but predicts the time evolution of the molecular dynamics simulation directly, rather than predicting the mapping from the geometry to potential energy. Following the protocol in Huang et al. (2022), we predict the atom position based on the velocity and coördinate 3000 steps prior. As shown in Table 5, SAKE achieves superior performance on 6 out of 8 systems, without leveraging spherical harmonics (as in TFN (Thomas et al., 2018) or SE(3)-TF (Fuchs et al., 2020)) or hand-coded edges (as in GMN (Huang et al., 2022)).

Table 4: Mean Squared Error (MSE) and inference time (ms)for charged particle dynamic system forecasting.

| Architecture | MSE | Inference time |
|---|---|---|
| SE(3)-TF (Fuchs et al., 2020) | 0.244 | 0.1346 |
| TFN (Thomas et al., 2018) | 0.155 | 0.0343 |
| GNN (Kipf and Welling, 2016) | 0.0107 | 0.0032 |
| EGNN (Satorras et al., 2021) | 0.0071 | 0.0062 |
| SAKE | 0.0049 | 0.0079 |
| SEGNN (Brandstetter et al., 2021) | **0.0043** | 0.0260 |

Table 5: Mean squarred error (MSE) ($10^{-2}$ Å$^2$) on MD17 trajectory forecast.

| | Aspirin | Benzene | Ethanol | Malonaldehyde | Naphthalene | Salicylic | Toluene | Uracil |
|---|---|---|---|---|---|---|---|---|
| TFN (Thomas et al., 2018) | $12.37_{\pm0.18}$ | $58.48_{\pm1.98}$ | $4.81_{\pm0.04}$ | $13.62_{\pm0.08}$ | $0.49_{\pm0.01}$ | $1.03_{\pm0.02}$ | $10.89_{\pm0.01}$ | $0.84_{\pm0.02}$ |
| SE(3)-TF (Fuchs et al., 2020) | $11.12_{\pm0.06}$ | $68.11_{\pm0.67}$ | $4.74_{\pm0.13}$ | $13.89_{\pm0.02}$ | $0.52_{\pm0.01}$ | $1.13_{\pm0.02}$ | $10.88_{\pm0.06}$ | $0.79_{\pm0.02}$ |
| EGNN (Satorras et al., 2022) | $14.41_{\pm0.15}$ | $62.40_{\pm0.53}$ | $4.64_{\pm0.01}$ | $13.64_{\pm0.01}$ | $0.47_{\pm0.02}$ | $1.02_{\pm0.02}$ | $11.78_{\pm0.07}$ | $0.64_{\pm0.01}$ |
| GMN (Huang et al., 2022) | $9.76_{\pm0.11}$ | $\mathbf{48.12}_{\pm0.40}$ | $4.63_{\pm0.01}$ | $\mathbf{12.82}_{\pm0.03}$ | $0.40_{\pm0.01}$ | $0.88_{\pm0.01}$ | $\mathbf{10.22}_{\pm0.08}$ | $0.59_{\pm0.01}$ |
| SAKE | $\mathbf{9.33}_{\pm0.02}$ | $137.20_{\pm0.06}$ | $\mathbf{4.63}_{\pm0.00}$ | $\mathbf{12.81}_{\pm0.03}$ | $\mathbf{0.38}_{\pm0.00}$ | $\mathbf{0.82}_{\pm0.01}$ | $10.98_{\pm0.01}$ | $\mathbf{0.53}_{\pm0.00}$ |

**Walking motion capture**   (CMU, 2003) has a higher system complexity and noise and is adopted to demonstrate SAKE's general capacity to forecast dynamic systems beyond microscopic scale. In this task, again closely following the experiment setting of Huang et al. (2022); Kipf et al. (2018), we predict the position of a walking person (subject 35 in CMU motion capture database (CMU, 2003)) based on their initial position. Again, we observe that SAKE outperform other models by a large margin (Table 6) and is significantly faster.

## 6.3 ABALATION STUDY

Table 7: SAKE performance on MD17-Aspirin (also see Table 1) with various components included (Y) or excluded (N).

| Spatial attention Eq. 6 | Semantic attention Eq. 10 | Speed and position update Eq. 12 | Energy RMSE (meV) | Force RMSE (meV/Å) |
|---|---|---|---|---|
| Y | Y | Y | $5.91^{5.92}_{5.88}$ | $8.09^{8.10}_{8.08}$ |
| N | Y | Y | $8.08^{8.09}_{8.05}$ | $17.48^{17.51}_{17.47}$ |
| Y | Y | N | $6.21^{6.23}_{6.20}$ | $10.31^{10.33}_{10.31}$ |
| N | Y | N | $8.15^{8.17}_{8.12}$ | $16.58^{16.61}_{16.57}$ |
| Y | N | Y | $7.88^{7.90}_{7.85}$ | $16.21^{16.22}_{16.19}$ |
| N | N | Y | $10.78^{10.79}_{10.75}$ | $17.90^{17.97}_{17.81}$ |
| Y | N | N | $6.29^{6.30}_{6.27}$ | $10.52^{10.54}_{10.50}$ |
| N | N | N | $8.21^{8.24}_{8.20}$ | $16.62^{16.64}_{16.60}$ |

To elucidate each component's contribution towards the final performance, we perform an ablation study on one of the most popular tasks studied so far—MD17 potential energy modeling (Section 6.1, Table 1). More specifically, we focus on the most complicated molecular system in the dataset, aspirin. We inspect three components proposed in the paper—*spatial attention*

Table 6: Walking motion capture performance.

| | GNN | EGNN Satorras et al., 2021 | GMN Huang et al., 2022 | SAKE |
|---|---|---|---|---|
| MAE | 67.3±1.1 | 59.1±2.1 | 43.9±1.1 | **14.59** ±1.6 |
| Epoch time | | | 5.66 s | **1.81** s |

(Equation 6), which we argue is the main novelty of the paper, as well as semantic attention (Equation 10) and velocity and position update (Equation 12). As is shown in Table 7, *spatial attention* improves the performance regardless of the rest of the configuration. To clearly demonstrate the utility of these auxiliary modules, we also add these components (except spatial attention) to the backbone of EGNN (Satorras et al., 2022), and summarize the results in Appendix Table 10—qualitatively, with these modules, EGNN achieves decent performance on par with SchNet (Schütt et al., 2017), with the performance gap comes primarily from its inability to conceive the node angular environments.

## 7 DISCUSSION

**Conclusions** In this paper we have introduced an invariant/equivariant functional form termed *spatial attention* that uses neurally parametrized linear combinations of edge vectors to equivariantly yet universally characterize node environments at a fraction of the cost of state-of-the-art equivariant approaches that makes use of spherical harmonics. Equipped with this spatial attention module, we use it to build a *spatial attention kinetic network* (SAKE) architecture which is permutationally, translationally, rotationally, and reflectionally equivariant. We have demonstrated the utility of this model in $n$-body physical modeling tasks ranging from potential energy prediction to dynamical system forecasting.

**Limitations** *Theoretical:* The universality condition is only discussed w.r.t. node geometry, without considering node embeddings; moreoever, the inequality condition in Theorem 1, although rarely violated in physical modeling (even when system is highly symmetrical), can be potentially over-restricting. We plan to generalize this framework to consider the expressive power of functions on the joint space of node embedding and geometry in future works. *Experimental:* Herein, apart from the novel functional form spatial attention (Equation 6), the rest of the architecture has not been thoroughly optimized and analyzed in the context of the growing design space of equivariant neural networks.

**Future directions** We plan to conduct more thorough experiments on (bio)molecular systems to explore the potential of SAKE in building general protein/small molecule force fields and enhanced sampling methods that could facilitate large-scale simulations useful in therapeutics and material discovery.

**Social impact and ethics statement**    This work provide an accurate and extremely efficient way to approximate properties and dynamics of molecular systems and physical states. It may advances research in a wide range of disciplines, including physics, chemistry, biochemistry, biophysics, and drug and material discovery. As with all molecular machine learning methods, negative implications may be possible if used in the design of explosives, toxins, chemical weapons, and overly addictive recreational narcotics.

**Reproducibility statement**    The software package containing the algorithm proposed here is distributed open source under MIT license. All necessary code, data, and details to reproduce the experiments can be found in Appendix Section 9.

**Funding**    Research reported in this publication was supported by the National Institute for General Medical Sciences of the National Institutes of Health under award numbers R01GM132386 and R01GM140090. YW acknowledges funding from NIH grant R01GM132386 and the Sloan Kettering Institute. JDC acknowledges funding from NIH grants R01GM132386 and R01GM140090.

**Disclaimer**    The content is solely the responsibility of the authors and does not necessarily represent the official views of the National Institutes of Health.

**Disclosures**    JDC is a current member of the Scientific Advisory Board of OpenEye Scientific Software, Redesign Science, Ventus Therapeutics, and Interline Therapeutics, and has equity interests in Redesign Science and Interline Therapeutics. The Chodera laboratory receives or has received funding from multiple sources, including the National Institutes of Health, the National Science Foundation, the Parker Institute for Cancer Immunotherapy, Relay Therapeutics, Entasis Therapeutics, Silicon Therapeutics, EMD Serono (Merck KGaA), AstraZeneca, Vir Biotechnology, Bayer, XtalPi, Interline Therapeutics, the Molecular Sciences Software Institute, the Starr Cancer Consortium, the Open Force Field Consortium, Cycle for Survival, a Louis V. Gerstner Young Investigator Award, and the Sloan Kettering Institute. A complete funding history for the Chodera lab can be found at `http://choderalab.org/funding`.

**Acknowledgments**    The authors would like to thank the ICLR reviewers for providing constructive feedback that substantially improved the quality and clarity of this manuscript. YW thanks Theofanis Karaletsos, Insitro, and Andrew D. White, University of Rochester, for useful discussions and Leo Klein, Freie Universität Berlin, for catching an embarrassing bug.

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

## 8  Proofs

### 8.1  Proof for Theorem 1

*Proof.* (Informal) An appropriate set of choices to satisfy the requirement is:

$f = I$, and $\lambda$ takes the set of $|\mathcal{N}(v)| + |\mathcal{N}(v)|^2$ elements:

$$\left\{ \lambda_i(h_{e_{u_j v}}) = [i = j] \right\} \cup \left\{ \lambda_{kl}(h_{e_{u_j v}}) = [k = j] - [l = j] \right\} \tag{14}$$

where $[\cdot]$ is the Iverson bracket. The first $|\mathcal{N}(v)|$-set of one-indexed functions give the node-to-neighbor distances whereas the second $|\mathcal{N}(v)|^2$-set of two-indexed functions give the distances among neighbors. This recovers the distance matrix $A_\mathcal{X}(v)$ among the node and its neighbors $A_\mathcal{X}(v)_{ii} = ||\mathbf{x}_v - \mathbf{x}_{u_i}||$ and $A_\mathcal{X}(v)_{ij} = ||\mathbf{x}_{u_j} - \mathbf{x}_{u_i}||, i \neq j, 1 \leq i, j \leq |\mathcal{N}(v)|$. As such, the relative positions of the node and its neighbors are uniquely defined up to E(n) symmetry (Havel, 1998). In other words, $\mathbf{x}_v$ and $\mathbf{x}_{u_i}$ can be uniquely embedded on an arbitrary coördinate system on which $g$ is evaluated. Since $\{\lambda\}$ is neural, by the universal approximation theorem (Hornik et al., 1989), $A_\mathcal{X}(v)$ can still be recovered, at least through recovering the set of $\{\lambda\}$ discussed above. Considering again the universal approximation of $\mu$, any function of the geometric node environment can be approximated by $\phi^{\mathrm{SA}}$. □

## 9  Detailed Methods

### 9.1  Code Availability

The corresponding software package and all scripts used to conduct the experiments in this paper are distributed open source under MIT license at: https://github.com/choderalab/sake This package can be installed via: `pip install sake-gnn`.

## 9.2 HARDWARE CONFIGURATION

All models are trained on NVIDIA Tesla V100 GPUs. Following the settings reported in the publications of baseline models, the inference time benchmark experiments (Section 6.2, 6.1) are done on NVIDIA GeForce GTX 1080 Ti GPU (For Table 1 and Table 6) and NVIDIA GeForce GTX 2080 Ti GPU (For Table 4).

## 9.3 ARCHITECTURE AND OPTIMIZATION DETAILS

One-layer feed-forward neural networks are used as $f_r$ in Equation 2 (edge update); two-layer feed-forward neural networks are used as $\phi^e$ in Equation 2 (edge update), $\phi^v$ in Equation 4 (node update), $\phi^{v \to \mathcal{V}}$ in Equation 12 (velocity update), and $\mu$ in Equation 6 (spatial attention). SiLU is used everywhere as activation, except in Equation 12 (velocity update) where the last activation function is chosen as $y = 2*\text{Sigmoid}(x)$ to constraint the velocity scaling to between 0 and 2 and in Equation 10 where CeLU is used before attention; additionally, $\tanh$ is applied on the additive part of Equation 12 to constraint it to between -1 and 1. 4 attention heads are used with $\gamma$ in Equation 10 spaced evenly between 0 and 5 Å. 50 RBF basis are used, spacedly evenly between 0 and 5 Å. All models are optimized with Adam optimizer. We summarize the hyperparameters used in these experiments in Table 8. All random seeds are fixed as 2666, as an homage to Bolaño (2004).

| Experiment | Depth | Width | Learning Rate | Epochs | Batch Size | L2 Regularization | Cutoff |
|---|---|---|---|---|---|---|---|
| MD17 (Table 1) Aspirin | 8 | 32 | $10^{-3*}$ | 5000 | 4 | $10^{-5}$ | 5.0 |
| MD17 (Table 1) Ethonal | 8 | 32 | $10^{-3*}$ | 5000 | 4 | $10^{-5}$ | 10.0 |
| MD17 (Table 1) Malonaldehyde | 8 | 64 | $10^{-3*}$ | 5000 | 4 | $10^{-5}$ | 10.0 |
| MD17 (Table 1) Naphthalene | 4 | 64 | $10^{-3*}$ | 5000 | 4 | $10^{-5}$ | 5.0 |
| MD17 (Table 1) Salicylic Acid | 6 | 64 | $10^{-3*}$ | 5000 | 4 | $10^{-5}$ | 5.0 |
| MD17 (Table 1) Toluene | 8 | 32 | $10^{-3*}$ | 5000 | 4 | $10^{-5}$ | 5.0 |
| MD17 (Table 1) Uracil | 8 | 64 | $10^{-3*}$ | 5000 | 4 | $10^{-5}$ | 5.0 |
| MD17 Trajectory Forcast (Table 5) | 2 | 8 | $10^{-3}$ | 1000 | 4 | $10^{-5}$ | |
| ISO17 (Table 2) | 6 | 64 | $10^{-3*}$ | 100 | 128 | $10^{-12}$ | |
| QM9 (Table 3) | 6 | 32 | $10^{-4*}$ | 5000 | 32 | $10^{-10}$ | |
| N-Body Forecast (Table 4) | 4 | 32 | $5*10^{-4}$ | 1000 | 100 | $10^{-12}$ | |

Table 8: Hyperparameters used in experiments (* A cosine warm up and annealing schedule is used, where the learning rate is gradually increased from $10^{-6}$ to the peak value in the first 10% epochs and decreased in the rest 90%.)

## 9.4 DATA AVAILABILITY

The source and details of benchmark datasets are summarized in Table.

| Experiment | License | Size | Split |
|---|---|---|---|
| MD17 http://quantum-machine.org/gdml/#datasets (Table 1) | | 8 Systems; 100K-1M snapshots | Random: 1K Train |
| ISO17 http://quantum-machine.org/datasets/(Table 2) | | 129 Molecules; 5000 snapshots | Fixed |
| QM9 http://quantum-machine.org/datasets/ | | 135k molecules | Fixed |
| N-Body Forecast [2](Table 4) | MIT | 5 particles | Fixed: 3K Train; 2K Valid; 2K Test |

Table 9: Dataset details.

# 10 ABALTION STUDY ON EGNN

# 11 BRIEF INTRODUCTION OF EQUIVARIANT NORMALIZING FLOWS

Normalizing flows (Rezende and Mohamed, 2016; Papamakarios et al., 2021) are a family of learnable bijections $f_{zx} : \mathcal{Z} \to \mathcal{X}$ that transform a tractable distribution on latent space $q_z(\mathbf{z})$ to another on the target space $\mathbf{x} = f_{zx}(\mathbf{z}; \theta)$ (dropping dependency on parameter thereafter) whose density can be analytically written as

$$\log q_x(\mathbf{x}) = \log q_z(\mathbf{z}) + \log \det |\frac{\partial f_{xz}(\mathbf{x})}{\partial \mathbf{x}}| \tag{15}$$

Table 10: EGNN (Satorras et al., 2022) performance on MD17-Aspirin (also see Table 1) with various components included (Y) or excluded (N).

| Distance smearing Eq. 7 | Semantic attention Eq. 10 | Position update Eq. 12 | Energy RMSE (meV) | Force RMSE (meV/Å) |
|---|---|---|---|---|
| Y | Y | Y | $25.92^{25.99}_{25.89}$ | $42.83^{42.86}_{42.78}$ |
| Y | Y | N | $16.02^{16.06}_{15.97}$ | $29.59^{29.62}_{29.55}$ |
| Y | N | Y | $31.57^{31.64}_{31.52}$ | $36.97^{37.01}_{36.94}$ |
| Y | N | N | $17.34^{17.39}_{17.31}$ | $33.85^{33.90}_{33.82}$ |
| N | Y | Y | $589.34^{590.44}_{588.11}$ | $217.15^{217.69}_{216.70}$ |
| N | Y | N | $588.95^{590.26}_{587.55}$ | $218.73^{218.97}_{218.64}$ |
| N | N | Y | $250.18^{251.17}_{249.43}$ | $538.68^{539.14}_{538.56}$ |
| N | N | N | $206.13^{206.71}_{205.79}$ | $882.32^{882.81}_{882.03}$ |

and conversely from a sample on the target space to the latent space $\mathbf{z} = f_{xz}(\mathbf{x}; \theta)$ whose likelihood is given by

$$\log p_z(\mathbf{z}) = \log p_x(\mathbf{x}) + \log \det \left| \frac{\partial f_{zx}(\mathbf{z})}{\partial \mathbf{z}} \right| \tag{16}$$

where $f_{zx} = f_{xz}^{-1}$. To close the gap between the intractable $p_x$ and the tractable $q_x$, so that one can sample on the target space efficiently, the flow is then trained by maximizing either Equation 15, if an unnormalized target distribution is given, or Equation 16, if samples are given.

The concept of *equivariant* normalizing flow was first introduced in Köhler et al. (2020); Rezende et al. (2019); Satorras et al. (2022), where they adopted the framework of continuous normalizing flow (Chen et al., 2018) to define the bijection $f_{zx}$ as

$$\mathbf{x} = \int_0^1 f'_{zx}(\mathbf{z}(t)) \, \mathrm{d}\, t \tag{17}$$

and restrict $f'_{zx}$ as E(n)-equivariant w.r.t. $\mathbf{z}$. Integration, or the sum over infinitely many equivariant functions, does not alter the equivariance. To numerically approximate this integration Chen et al. (2019) involves evaluating $f'_{zx}$ multiple times and is therefore expensive.

## 12 EQUIVARIANT EXACT LIKELIHOOD SAMPLING WITH SAKE FLOW.

Current equivariant normalizing flow models (Satorras et al., 2022; Köhler et al., 2020) (briefly reviewed in Appendix Section 11), relies on ODE-based numerical integration(Chen et al., 2019), are computationally expensive. We propose a much simpler invertible flow model that uses our SAKE model, termed SAKE Flow. First, following a scheme introduced in Huang et al. (2020) (in which it is argued that with flexible enough kernels Equation 17 could be approximated arbitrarily well), we extend the space $\mathcal{X}$ with an auxiliary space $\mathcal{A}$. Correspondingly, we extend the tractable distribution to be $q(\mathbf{z}, \mathbf{a}) = q_z(\mathbf{z})q_a(\mathbf{a})$ and the target distribution to be $p(\mathbf{x}, \mathbf{a}) = p_x(\mathbf{z})q_a(\mathbf{a})$. We then change the problem statement of Equation 16 to: find a parametrized function $f_{zx}(\mathbf{z}, \mathbf{a}; \theta) = f_{xz}^{-1}(\mathbf{x}, \mathbf{a}; \theta)$ that is a bijection on the space $\mathcal{A} \times \mathcal{X}$ to maximize the joint likelihood:

$$\hat{\theta} = \operatorname{argmax} \mathbb{E}_{\mathbf{a} \sim q_a}[\log p(\mathbf{x}, \mathbf{a}))] \tag{18}$$

which is, up to a constant, an evidence lower bound for $\mathbf{x}$:

$$\log p_x(\mathbf{x}) \tag{19}$$
$$= \mathbb{E}_{q_a}[\log p(\mathbf{x}, \mathbf{a})] + D_{\mathrm{KL}}[q(\mathbf{a})||p(\mathbf{a}|\mathbf{x})] + H_a \tag{20}$$
$$\geq \mathbb{E}_{q_a}[\log p(\mathbf{x}, \mathbf{a})] + H_a, \tag{21}$$

The equality holds when the (non-negative) Kullback–Leibler divergence between the tractable distribution $q_a$ and the conditional distribution given samples $p(\mathbf{a}|\mathbf{x})$ is zero. As such, an unbiased estimate of the marginal likelihood can be given by

$$\log \hat{p}(\mathbf{x}) = \log p(\mathbf{x}, \mathbf{a}) - \log q_a(\mathbf{a}). \tag{22}$$

Similar to Huang et al. (2020); Dinh et al. (2017), we define $f_{zx}(\mathbf{z}, \mathbf{a}, \theta)$ as a series of alternating affine coupling:

$$g^{z\to a/a\to z}(\mathbf{z}, \mathbf{a}) = \mathbf{z}, \exp(S^{z\to a/a\to z}(\mathbf{z})) \odot \mathbf{a} + T^{z\to a/a\to z}, \tag{23}$$

with analytic inverse. Composing these transformations $g^{z\to a} \circ g^{a\to z} \circ ...g^{z\to a} \circ g^{a\to z}$, we get our bijection $f_{zx}(\mathbf{z}, \mathbf{a}; \theta)$.

Now, it only remains to define the structure of the translation functions $\{T\}$ and scaling functions $\{S\}$. As is outlined in Satorras et al. (2022), if $f_{zx}$ is *translation-invariant*, it is impossible to have $\int p_x(\mathbf{x})\, \mathrm{d}\, x = 1$, so we drop the translation invariance requirement and design $f_{zx}$, as well as the composing $\{T, S\}$, to be only rotation and reflection equivariant. Correspondingly, we require all tractable distributions to be confined on a $(|\mathcal{V}| - 1)n$-dimensional subspace with $\mathbf{0}$ gravity center. (Note that this reduces the symmetry group we work on from E(n) to SO(n).) A valid choice both for $q_z(\mathbf{z})$ and $q_a(\mathbf{a})$ with is a *centered Gaussian* distribution (Satorras et al., 2022):

$$p(\mathbf{z}) = \frac{1}{(2\pi)^{|\mathcal{V}-1|n/2}} \exp(-\frac{1}{2}||\mathbf{z}||^2) \tag{24}$$

with $\sum_{v\in\mathcal{V}} \mathbf{x}_v = \mathbf{0}$. Moreover, to keep the gravity center at $\mathbf{0}$, we require that $\{T, S\}$ shall not change the gravity center. A general recipe to construct such $\{T, S\}$ is to use the *equivariant* and *invariant* outputs of SAKE to parametrize $T$ and $S$ respectively.

$$h, \mathbf{x} = \mathrm{SAKEModel}(h, \mathbf{x}); \mathbf{x} = \mathbf{x} - \mathrm{MEAN}(\mathbf{x}); h = \exp(\mathrm{MEAN}(h));$$
$$T(\mathbf{z}) = \mathbf{z} + \mathbf{x}; S(\mathbf{z}) = h * \mathbf{z} \tag{25}$$

To preserve the center of gravity, we center the translation to have zero center of gravity and enforce the same scalar to be used across particles as the scaling factor.

