# OpenReview forum: "Spatial Attention Kinetic Networks with E(n)-Equivariance"
_ICLR.cc/2023/Conference — ICLR 2023 poster_

### Official Review · Reviewer_g5jj · 2022-10-23

**Confidence:** 5
**Correctness:** 3
**Technical Novelty And Significance:** 2
**Empirical Novelty And Significance:** 3
**Recommendation:** 6

**Clarity, Quality, Novelty And Reproducibility:**

- Overall, the paper is well-written and easy to follow. There are some technical details missing which makes it hard to read occasionally.
- The novelty of the spatial attention architecture seems marginal. It extends the equivariant update in Eq. 4 of the EGNN paper from node coordinates to node features.
- The authors provided the code to reproduce the results in anonymous repo.


**Strength And Weaknesses:**

Strength:
- The introduced spatial attention improves the expressiveness of the EGNN and preserves the E(n)-Equivariance of the original model.
- The paper conducts extensive experiments to demonstrate the improved performance in invariant property prediction, equivariant property prediction, and density estimation in equivariant normalizing flows.
- Model inference is significantly faster than architectures based on spherical harmonics.
- Detailed ablation study is performed to compare SAKE and EGNN.

Weakness:

- The ablation results are confusing. EGNN is significantly worse than SAKE (for Aspirin, SAKE has MAE of 9.9 and EGNN has an MAE of 298.05). It looks like the EGNN is not properly trained. Further, the authors need to provide more detailed descriptions for the ablation study. What does “No update” mean?
- What is the equivariant function f in Eq. 6? I cannot find a detailed explanation of the function f in the paper. Is it an identity function or other equivariant function?
- The Proof 8.1 only considers the special case of f = I. Will Theorem 1 still hold if f is not the identity function?
- Some inference time comparisons are incomplete. In Table 1, the inference time of SchNet, sGDML, PaiNN is missing. In Table 5, the inference time of GNN, EGNN is also missing. How much overhead does the attention mechanism introduce compared with GNN and EGNN?
- In QM9, the model performs less competitively in some intensive property prediction tasks. The authors hypothesize that it can be mitigated by size-invariant pooling functions. This should be a straightforward improvement by using a mean pooling rather than sum pooling. I suggest providing additional results using the mean pooling.


**Summary Of The Paper:**

This paper introduces spatial attention into E(n)-equivariant neural networks to describe the node environments for the modeling of many-body systems. The architecture builts on the EGNN introduced by Satorras et al. and introduces spatial attention to improve the performance in several  invariant and equivariant prediction tasks.

**Summary Of The Review:**

Overall, the paper introduces a novel equivariant spatial attention layer, extending the EGNN architecture. The authors demonstrated improved performance in a broad range of tasks. However, the novelty of the spatial attention layer seems marginal. More detailed ablation study is needed to clarify the usefulness of the introduced spatial attention layer.

---

> ### Author Response · Authors · 2022-11-07
> **Clarification on the equivariant function requirement; EGNN ablation performance**
>
> Thank you, Reviewer `g5jj`, for your constructive feedback, and for recognizing the extensiveness of experiments which illustrated the performance and efficiency of the model.
>
> We would like to provide some initial comments regarding your concerns, while we prepare more numerical experiments to elucidate the efficacy of our model:
>
> > The ablation results are confusing. EGNN is significantly worse than SAKE.
>
> The problem with the original for EGNN is twofold: firstly EGNN does not use distance smearing, which, though not improving expressiveness, has been shown to improve trainability (see SchNet by Schütt 2017). When equipped with RBF smearing, EGNN essentially has similar performance with SchNet (see the row “EGNN + RBF” in comparison with SchNet performance in Table 1). Secondly, neither SchNet nor EGNN encodes angular environment, which make them unsuitable for a detailed molecular dynamics energy landscape modeling, which has motivated our model. We have also clarified the confusing “no update” column in the ablation study.
>
> > What is the equivariant function f in Eq. 6? The Proof 8.1 only considers the special case of f = I. Will Theorem 1 still hold if f is not the identity function?
>
> In Theorem 1 we argue the _existence_ of such equivariant function $f \in \mathcal{X} \rightarrow \mathcal{X}$,  and show in Proof 8.1 that even the identity function can satisfy such requirement. By the same logic of Proof 8.1, any distance matrix-preserving equivariant functions namely combinations of rotations, reflections, and translation, also suffice.
>
> > Some inference time comparisons are incomplete. How much overhead does the attention mechanism introduce compared with GNN and EGNN?
>
> In order to prevent inaccurate reporting on other models’ inference time, we only report inference time from other papers and adopt their exact experimental conditions. In Table 4 we show that SAKE is only slightly slower than EGNN.

---

> ### Author Response · Authors · 2022-11-18
> **Rewritten ablation study**
>
> Thanks again, Reviewer `g5jj`, for your thorough and helpful feedback. We have significantly revised the paper to address many of your concerns. Please see [the comment for all reviewers above.](https://openreview.net/forum?id=3DIpIf3wQMC&noteId=ZUdGA6gH6I) More specifically:
>
> > The ablation results are confusing. EGNN is significantly worse than SAKE (for Aspirin, SAKE has MAE of 9.9 and EGNN has an MAE of 298.05). It looks like the EGNN is not properly trained.
>
> We have reworked the ablation study session. To explain EGNN’s underwhelming performance, various components were also added to EGNN to the point where it conceptually resembles a SchNet model. It turns out that without the distance smearing module (namely RBF), EGNN is indeed difficult to train. With RBF added to EGNN, the performance difference between EGNN and SAKE lies primarily in the module to realize angular environments.
>
> > Some inference time comparisons are incomplete.
>
> As shown in the reworked manuscript, we only report published inference wall time from their original publication to avoid possible inaccurate report due to suboptimal software configuration. We did, however, make effort to ensure the hardware conditions match among experiments.

---

> > ### Comment · Reviewer_g5jj · 2022-12-07
> > **Thanks for your reply**
> >
> > Sorry for being slow in my replies. I've gone through the revised paper and comments from other reviewers. The focus on spatial attention and new ablation improved the clarity of the paper. The authors show that the spatial attention improves the performance over EGNN in broad range of tasks. The novelty of the approach is still limited in my opinion. But I now changed my score to 6 for the broad usage of the architecture improvement.

---

> > > ### Author Response · Authors · 2022-12-08
> > > **Thank you!**
> > >
> > > Thanks a lot for reading the revised manuscript and for acknowledging our improvements.

---

> ### Author Response · Authors · 2022-11-25
> **Thank you for your feedback.**
>
> Thank you, Reviewer `g5jj`, for your feedback! We believe that we have addressed the majority of it. Please let us know if there is anything we can do to further improve this paper!

---

> ### Author Response · Authors · 2022-11-29
> **Looking forward to your comment**
>
> Hi Reviewer `g5jj`,
>
> We have revised the paper to incorporate your suggestions and concerns, specifically in terms of the ablation study and the theoretical framework. Please let us know if you have further comments and/or suggestions.
>
> Thanks a lot!

---

### Official Review · Reviewer_u1rJ · 2022-10-30

**Confidence:** 4
**Correctness:** 3
**Technical Novelty And Significance:** 3
**Empirical Novelty And Significance:** 3
**Recommendation:** 8

**Clarity, Quality, Novelty And Reproducibility:**

The writing of the paper could be better but is generally quite clear. To the best of my knowledge, the proposed method is novel (although it relies very heavily on two papers). It seems to have major advantages in terms of scalability and generally returns competitive performance across a range of tasks.

**Strength And Weaknesses:**

Strengths:
- The paper addresses an important problem. The method is well-motivated and quite simple and elegant.
- Experimental results are quite encouraging. The presented approach is competitive with various methods that were proposed recently -- although it is not the best across the board -- but has a significant advantage in terms of run-time.
- I think the formulation also suggests approaches to optimize for the architecture, which can improve the experimental results further.

Weaknesses:
- I think the writing of the paper is a little bit clunky.
- The novelty of the method is somewhat limited. But I think the experimental results make the case for it as far as I am concerned.
- The code is not properly anonymized.

Minor comments:
- At the beginning of page 3, \mathcal{H} is D dimensional, but at the beginning of section 4, it is written to be C dimensional. Might want to remove this inconsistency.
- Please re-write the abstract. For instance, the first line is too long and seems to have some punctuation missing.

Some typos
- Page 2, line 2: invaraince --> invariance.
- Page 2, line 2 in paragraph 2: "keep track" --> keeps track.
- Page 3, first paragraph of section 3: "Compared to SAKE, these models are not capable in equivariant modeling" --> "Compared to SAKE, these models are not capable of equivariant modeling".
- Page 4, below equation 6: "function that operate on the edge vector" --> "function that operates on the edge vector"
- Page 4, below remark 1: "With all local degrees of freedom are incorporated in spatial attention" --> "With all local degrees of freedom incorporated in spatial attention,"
- Page 5, above section 6: "In relation to spherical harmonics-based models." --> "Relation to spherical harmonics-based models."
- Page 7, section 6.2: "model can predict the evolving of a physical system sufficiently long after initial conditions" --> "model can predict the evolution of a physical system sufficiently long after initial conditions"
- Page 8, next to table 5: "invaraince" --> invariance.
- Page 8, next to table 5: "as the composing" --> "as the composition"
- Page 8: "Note that this reduce" --> "Note that this reduces"

**Summary Of The Paper:**

Some of the earlier successful methods for physical modelling (like Schutt et al., 2017 and Bartok et. al, 2013) are unable to fully account for the geometry of the system. To model particle arrangements, they rely on using radial information (which captures inter-particle distances). It should be straightforward to see that such methods can't fully capture the geometry of node environments. For instance, one might have particular arrangements which are quite different relative to each other, even when the distance from a node remains unaltered.  More recent methods use a truncated series (resembling a multipole expansion) that uses spherical harmonics to encode higher-order interactions between particles -- and then incorporate it in a message-passing formulation. Such methods have proven to be quite powerful and have been developing quite rapidly over the past few years. However, they do tend to be somewhat expensive computationally. This paper attempts to address this by proposing a method (SAKE) that relies on a simple attention-based formulation that uses linear combinations of edge vectors while being equivariant. More specifically, SAKE uses the _norm_ of linear combinations of edge features to specify the node environment. The method and some of the contributions are summarized below:

The background on equivariance and a high-level description of the message-passing formalism is provided in sections 2.1 and 2.2. The problem is formulated as implementing a function f_\theta: \mathcal{X} x \mathcal{H} \mapsto \mathcal{X} x \mathcal{H}. The space \mathcal{X} x \mathcal{H} forms a joint space of n-dimensional coordinates, and D-dimensional semantic embeddings. The approach that is most similar in terms of applicability and import is that of Satorras et al. The framework for spatial attention is defined in section 4. It uses an equivariant function on the edge embeddings. These embeddings are used to generate attention weights \lambda. The authors suggest taking N_{\lambda} such linear combinations, computing their norms and then further concatenating them (multiple heads), and mapping it to the \mathcal{H} space (by function \mu). Both the \lambda and \mu functions are modelled via MLPs. This simple spatial attention module is E(n) invariant. It is also universal for E(n) invariant functions, as encapsulated in theorem 1. The edge embedding is computed by borrowing ideas from Schutt et. al. and Satorras et al. (equation 7). In addition, there are two more attention modules to promote anisotropy (described on page 5). Finally, there is a similar term as in Satorras et al. that models updating a fictitious velocity (eq 10 and 11).

SAKE is equivariant on the geometric space and invariant on the embedding space. The experiments are divided into three parts. For invariant modelling, the authors consider standard tasks on MD17, ISO17, and QM9.  On MD17, the approach is competitive, and beats all but one of the baselines, but with a significant time gain. While only three baselines are considered for ISO17, SAKE is reported to be better than the competition, while again reportedly faster. Then there are two equivariant tasks (charged N-body and walking motion) where a similar trend is reported. Finally, the method is used to describe a normalizing flow method and is shown to be competitive with the approaches of Kohler et al. and Satorras et al. 2022.


**Summary Of The Review:**

See the above summary.

---

> ### Author Response · Authors · 2022-11-07
> **Thank you for your constructive feedback!**
>
> We thank Reviewer `u1rJ` for their thorough read of the paper and constructive feedback and for pointing out our encouraging experimental results. Additionally, we thank you for combing through our paper and paying attention to the details. We have addressed these comments and will keep improving the manuscript for readability and clarity.  In terms of the code anonymity, we were not aware that the open source anonymizing software would not handle the GitHub license. We have fixed this issue.

---

> > ### Comment · Reviewer_u1rJ · 2022-11-17
> > **Thanks**
> >
> > Thanks for fixing the issue! I will go over the updates and comment if any new questions arise.

---

> > > ### Author Response · Authors · 2022-11-18
> > > **Many thanks and see comment above**
> > >
> > > Many thanks!
> > >
> > > We have reworked our manuscript to address concerns from the reviewers. Please see [the comment above](https://openreview.net/forum?id=3DIpIf3wQMC&noteId=ZUdGA6gH6I) and the updated manuscript. Thanks again for your constructive feedback.

---

> > > > ### Comment · Reviewer_u1rJ · 2022-11-29
> > > > **Thanks for the revisions**
> > > >
> > > > Sorry it took me time to go over the revised paper and updates. I am satisfied with these updates, and they do improve on the general clarity -- a concern I kind of hinted at. I think the paper as it stands now addresses some other concerns raised. While I already vote for acceptance, I would increase my score by one notch as I see value in the paper being published.
> > > >
> > > > Regards

---

> > > > > ### Author Response · Authors · 2022-11-29
> > > > > **Thank you!**
> > > > >
> > > > > Many thanks for your feedback and encouraging comment! We will keep revising and refining the paper in terms of both experiments and representation.

---

### Official Review · Reviewer_ULzV · 2022-11-01

**Confidence:** 4
**Correctness:** 3
**Technical Novelty And Significance:** 2
**Empirical Novelty And Significance:** 2
**Recommendation:** 6

**Clarity, Quality, Novelty And Reproducibility:**

+ This paper is overall well written.

- The novelty is marginal and mainly lies in the proposal of spatial attention mechanism (Eq.6), which as mentioned above is over-restricted compared to recent methods.

- The results of EGNN in the table in Section 11 are much worse than other methods, which seems quite weird and unconvincing.


**Strength And Weaknesses:**

Strengths:

1. The design of spatial attention in Eq. (5) is well motivated and novel. As illustrated in the proof of Theorem 1, if certain condition meets, the spatial attention is able to recover the local environment of each node. The proof of Theorem 1 is interesting by recovering the distance matrix.

2. The part of Related Work is well organized, providing a good guidance of understanding of how recent progress is made.

3. The experimental comparison with various methods have been conducted.

Weaknesses:

1. The condition $h_{e_{vu_i}}\neq h_{e_{vu_j}}$ in Theorem 1 is over-restricted and easily broken when two neighbor nodes share the same embedding (the same node features and local context). The authors claimed that the inequality holds when the system is not strictly symmetrical. Unfortunately, many practical molecular systems are symmetric such as benzene and many other materials such as crystals.

2. How to build a permutation-invariant and E(n)-invariant (or equivariant) function has been discussed in [A], which is not cited, unfortunately. In [A], Propositions 10-11 have presented the universal form of message passing from local neighbors up to the O(n) equivariant. More importantly, the results by [A] does not require the condition $h_{e_{vu_i}}\neq h_{e_{vu_j}}$ that is indispensable in this paper. The authors are suggested to compare the difference with [A] which seems more elegant.

3. Moreover, the proof of Theorem 1 only shows the results on invariant function other than equivariant function. In [B] and its citated references, the universal form of equivariant message passing can be derived from TFN and GemNet, both of which again do not require any restriction of node embedding.

4. It is unclear why "The distinctness condition corresponds to the full-rank requirement in GMN".

5. The authors have shown experimental comparisons on MD17 and QM9, which are already well explored, and the results of the proposed method are not clearly superior to other methods. The authors are strongly suggested to conduct performance on other recent and more challenging tasks, such as the trajectory prediction setting of MD17 and the benchmark OC20-22.

6. The ablation studies are insufficient and unclear to justify the contribution of this paper. It seems from Algorithm 1 that the main novelty lies in the proposal of spatial attention Eq. 6 along with other minor techniques. The authors should detail how each proposed component affects the performance and provide necessary analyses.




[A] Scalars are universal: Equivariant machine learning, structured like classical physics, NIPS 2021.

[B] GemNet: Universal Directional Graph Neural Networks for Molecules, NIPS 2021.

**Summary Of The Paper:**

This paper exploits several kinds of attention mechanisms (Euclidean and semantic attention, spatial attention) upon EGNN [6]. Necessary theoretical claims have been made to justify the expressivity of the proposed spatial attention. Experiments are conducted on invariant tasks and equivariant tasks.

**Summary Of The Review:**

The authors are tackling a variable problem on enhancing the expressivity of equivariant GNNs on physical modeling. However, given the limited novelty, technicality, and insufficient ablations, I suggest weak reject for its current version.

---

> ### Author Response · Authors · 2022-11-07
> **Clarification on theoretical framework; more experimental evidence under work**
>
> Many thanks, Reviewer `ULzV`, for your thorough and constructive feedback, and for your encouraging comments on the simplicity of our theoretical framework and thoroughness of experimental benchmark.
>
> Per your suggestions, we are currently working on expanding the empirical benchmark to OC20-22 systems. In the ablation study (Section 11), we have already shown that the spatial attention mechanism contributes most to the performance whereas the contribution from other tricks are minimal. We nonetheless plan to expand this ablation study section to include an even more thorough comparison. We will share the updated numerical experiments with reviewers soon.
>
> Meanwhile, we would like to address some of your concerns in terms of the theoretical representation of this paper.
>
> > The condition $h_{e_{u_{vu_i}}} \neq h_{e_{u_{vu_j}}}$ Theorem 1 is over-restricted and easily broken when two neighbor nodes share the same embedding. Unfortunately, many practical molecular systems are symmetric such as benzene and many other materials such as crystals.
>
> Note that the edge embedding is iteratively updated by the geometric environment. So in physical modeling, any small distortion resulted from the vibration of the system is going to break the symmetry and thus allow this condition to hold. The benzene molecule you have proposed is a perfect example---it happens to be included in an older version of MD17 dataset, on which we have trained a model and observed all distinct embeddings among edges.
>
> > How to build a permutation-invariant and E(n)-invariant (or equivariant) function has been discussed in [A], which is not cited, unfortunately.
>
> Thank you for pointing this paper to us. It provided an elegant framework for constructing general invariant and equivariant functions on E(n). Our work could be regarded as a realization of its Propositions 10-11, albeit we use norm of parametrized linear combinations of these representations, rather than direct dot products, as scalarization. We have added sections of discussion to address the relationship between our work and theirs.
>
> > Moreover, the proof of Theorem 1 only shows the results on invariant function other than equivariant function.
>
> We have discussed the equivariant universal approximative properties in Remark 2. We realize that the way we arrange these to findings make it difficult for readers to connect these two pieces. We have added cross references to connect these two statements.

---

> > ### Author Response · Authors · 2022-11-18
> > **New experiments on MD17 forecast; rewritten theoretical section**
> >
> > Many thanks again, Reviewer `ULzV`, for your constructive feedback. As shown in [the comment above](https://openreview.net/forum?id=3DIpIf3wQMC&noteId=ZUdGA6gH6I "the comment above"), we have restructured the paper to incorporate many of your suggestions. For instance:
> >
> > > Moreover, the proof of Theorem 1 only shows the results on invariant function other than equivariant function.
> >
> > We have reworked the theoretical sections to make the link between invariant and equivariant universality clearer.
> >
> > > The ablation studies are insufficient and unclear to justify the contribution of this paper. It seems from Algorithm 1 that the main novelty lies in the proposal of spatial attention Eq. 6 along with other minor techniques. The authors should detail how each proposed component affects the performance and provide necessary analyses.
> >
> > We have rewritten the ablation studies section to clearly separate the contributions of various modules. Furthermore, we removed several “minor techniques” from the paper to focus extensively on spatial attention.

---

> ### Author Response · Authors · 2022-11-25
> **Thanks again for your feedback.**
>
> Thanks again, Reviewer `ULzV`, for your feedback. We believe that we have addressed the majority of your concerns. Could you please let us know if there are further question and/or concerns on this work? Thank you!

---

> ### Author Response · Authors · 2022-11-29
> **Looking forward to your comments**
>
> Hi Reviewer `ULzV`,
>
> Thanks to your feedback, we have revised the paper. We believe that the majority of your concerns are addressed. I would look forward to hearing your further comments and suggestions.
>
> Thanks again!

---

> > ### Comment · Reviewer_ULzV · 2022-12-07
> > **Score increased**
> >
> > I have checked the revised copy and am willing to increase the score from 5 to 6.

---

> > > ### Author Response · Authors · 2022-12-08
> > > **Thank you!**
> > >
> > > Many thanks for reading the revised copy and for increasing the score.

---

### Official Review · Reviewer_YAqW · 2022-11-01

**Confidence:** 4
**Correctness:** 4
**Technical Novelty And Significance:** 2
**Empirical Novelty And Significance:** 3
**Recommendation:** 6

**Clarity, Quality, Novelty And Reproducibility:**

Clarity: There appear to be many inaccuracies in the formulae. This makes it very hard to understand the details of the proposed method and follow the proofs.
- Above equation (6), the type $\phi: \mathcal X \times \mathcal H \to \mathcal H$ does not seem consistent with definition (6), which is more like $\phi: (\mathcal X \times \mathcal H)^{\mathcal N(v)} \to \mathcal H$.
- Similarly, $\mu: N_\lambda \to \mathcal H$ should probably be $\mu: \mathbb R^{N_\lambda} \to \mathcal H$.
- In theorem 1, the function $g$ should take a set of neighbours as input, not a single neighbour.
- As stated above, the definition of $\phi^{DPS}$ is not clear.
- In section 8.2, $\lambda_{u, i}$ in the definition of $\lambda_+$ and $\lambda_-$ should be $\lambda_{q, i}$

Quality: I have serious concerns about the correctness of both theoretical claims.

Novelty: The method is a small variation of pre-existing work.

Reproducibility: The source code is provided.

**Strength And Weaknesses:**

## Strength
- The method is evaluated on a wide variety of tasks.
- The method performs competitively.

## Weakness
- The core spatial attention method is a small variation of / combination of pre-existing methods.

### Representation of theorem 1
I think that theorem 1 is misrepresented in the paper. The authors write
> spatial attention is capable of universally approximating any functions defined on local node environment while preserving E(n)-invaraince/equivariance in arbitrary n-dimensional space.

I would interpret this as saying that any invariant/equivariant function from the neighbourhood features and coordinates can be represented by spatial attention. I would thus expect a theorem like:

> Given $N$ neighbours, $C$-dimensional features and $n$ spatial dimensions, let $V \subset \mathbb R^{C^N}$ be the subset of the edge features where all edges have distinct values. Then for any function $g : V \times \mathbb R^{n^N} \times \mathbb R^n \to \mathbb R$, that is invariant under permutations and $E(n)$, there is a spatial attention layer that matches $g$ arbitrarily well on all inputs.

However, this is not what is shown in the proof. The proof states that for **one given value of edge features** $\mathbb R^{C^N}$, and one invariant function of the coordinates $g: \mathbb R^{n^N} \times \mathbb R^n \to \mathbb R$, there is a corresponding spatial attention layer. The spatial attention is thus not shown to be universally express any function from the node neighbourhood with node features. Other than the paper suggests, the theorem thus also has no implication about whether the SAKE network can represent any equivariant function of pointclouds with node features. I think the claim that the layer is a universal approximator is thus incorrect.

Like theorem 1, I think that remark 3 wrongly claims universality.

### Remark 2
The DPS method is not so clearly defined, but for equation $\phi^{DPS}(v; \lambda_k, \lambda_q)=(\phi^{SA}(v; \lambda_+)^2 - \phi^{SA}(v; \lambda_-)^2)/2$ (Sec 8.2) to make sense, it seems like:

$\phi^{DPS}(v; \lambda_k \lambda_q)_i = \sum\_{uu'}\lambda\_{q,i,u}\lambda\_{k,i,u'} \langle f\_u , f\_{u'} \rangle$

where $f_u=f(e_{uv})\in \mathbb R^n$ and $\lambda\_{q,i,u}=\lambda\_{q,i}(h\_{e\_{uv}}) \in \mathbb R$ is the intended defintion.

If so, the claim in Sec 8.2 that with orthogonal edges $\phi^{DPS}(v)=0$ is clearly false. For example, if there is one neighbour with edge $e$, $N_\lambda=1, \lambda=1, f(e)=e, \mu(c)=c$, then it looks like $\phi^{DPS}(v)=||e||^2$. Thus the claim that the spatial attention is more expressive than DPS, doesn't appear correct.

A forteriori, it appears to me that a construction similar to the proof in Sec 8.1 can be used for $\phi^{DPS}$ to show that it is just as universal as the spatial attention method as defined in Theorem 1. Appropriately choosing $\lambda$, one can construct a list of all inner products $\langle x_v - x_u , x_v - x_{u'} \rangle$, from which all pairwise distances between the neighbours can be constructed via $||x_u - x_{u'}||^2=||x_v - x_{u}||^2+||x_v - x_{u'}||^2 - 2 \langle x_v- x_u, x_v - x_{u'} \rangle$. Subsequently, the same argument  as made in sec 8.1 can be made to show that this construction is universal as defined in theorem 1.

Also with a definition of $\phi^{DSA}$ that's more similar to conventional attention, like $\phi^{DPS}(v; \lambda_k \lambda_q)_{uu'} = \sum\_{i}\lambda\_{q,i,u}\lambda\_{k,i,u'} \langle f\_u , f\_{u'} \rangle$, choosing $\lambda=1,f\_u=e\_{vu}$, it is universal as defined in theorem 1.

In conclusion, regardless of how exactly $\phi^{DSA}$ is defined, it doesn't appear any less expressive than the the proposed spatial attention method. I thus don't think Remark 2 is correct.

## Typo
- p2, first paragraph "invaraince"


**Summary Of The Paper:**

The paper proposes an E(n) equivariant network acting on pointclouds with edges. The method is simple to compute and shows competitive performance on various tasks.

The authors claim that their attention method is universal, and that it is more expressive than "dot product scalarization" attention used in prior wok.

**Summary Of The Review:**

I have serious concerns about both theoretical claims in this paper: that the spatial attention mechanism is universal, and that it is more expressive than different attentional methods. I thus recommend rejection of this paper.

---
After their revision, the theoretical issues are fixed. I now think the paper is a valuable contribution to the literature on equivariant pointcloud networks and should be accepted. I weakly recommend acceptance, as the novelty is still limited.

---

> ### Author Response · Authors · 2022-11-07
> **Clarification on the scope of Theorem 1.**
>
> Thank you, Reviewer `YAqW`, for your constructive feed back. We would like to first address your main concern regarding Theorem 1.
>
> > I think that theorem 1 is misrepresented in the paper. I would interpret this as saying that any invariant/equivariant function from the neighbourhood features and coordinates can be represented by spatial attention. I would thus expect a theorem like:
>
> We would like to clarify that we have been concerning with the universality function $g: \mathcal{X} \rightarrow \mathbb{R}$, i.e. that maps the geometry space to scalars. We had made this clear in the paper and added further clarification. Reviewer `YAqW` _expects_ to see the universality discussed on the joint space between node features and geometric environments, which is out of the scope of our paper. Also, regardless of geometry, GNNs that operate on neighbors’ node features are in general not universal.
>
> > In conclusion, regardless of how exactly $\phi^\operatorname{DSA}$ is defined, it doesn't appear any less expressive than the the proposed spatial attention method. I thus don't think Remark 2 is correct.
>
> Thank you for pointing this out. We realized that if one included self dot product of edge equivariant representations, $\phi^\operatorname{DSA}$ can indeed be rendered as universal. We have deleted this remark from the manuscript and refer the readers to the experimental evidence to justify the empirical superiority of our method.

---

> > ### Comment · Reviewer_YAqW · 2022-11-09
> > **Theorem 1 still misrepresented**
> >
> > Dear authors,
> >
> > Thank you for understanding my concerns regarding Remark 2.
> > Regarding theorem 1, could you elaborate where you have clarified the notion of universality used in this paper. In the current version, I still think the universality properties are misrepresented. Also, as the prior works are apparently universal in the same way, that should be clarified in the paper.

---

> > > ### Author Response · Authors · 2022-11-09
> > > **further clarification: universal function operates only on the geometry**
> > >
> > > Many thanks again, Reviewer `YAqW`, on helping us shape the clear representation of this argument. We have made these modifications to address your concern:
> > >
> > > 1. We have rewritten the E(n)-invariant function to be $g(\mathbf{x} | h)$ rather than $g(\mathbf{x}, h)$ to emphasize that the embedding is a condition, not part of the domain.
> > >
> > > 2. We have added clarification texts right under the theorem to make it completely clear that we do not deal with node embeddings in this paper and to redirect readers to GNN representation power literature when it comes to node embeddings.
> > >
> > > 3. In the Discussion-Limitations section, we added a _Theoretical_ limitation statement that makes this limitation completely transparent as well as the potential over-restricting inequality condition mentioned by Reviewer `ULzV`.
> > >
> > > We appreciate your thorough read of this paper and insights on the accurate description of our framework. We plan to include more experimental results to compensate for the simplification of the theoretical section of this paper.

---

> > > > ### Comment · Reviewer_YAqW · 2022-11-24
> > > > **Issues resolved**
> > > >
> > > > I thank the authors for fixing the issues with the theory. I now recommend acceptance of the paper, as it's sound, relatively straightforward to implement and is shown competitive on a large number of datasets.

---

> > > > > ### Author Response · Authors · 2022-11-24
> > > > > **Thank you!**
> > > > >
> > > > > Thank you again, Reviewer `YAqW`, for your feedback!

---

> > > ### Author Response · Authors · 2022-11-18
> > > **further clarification**
> > >
> > > Hi Reviewer `YAqW`,
> > >
> > > Thank you again for helping us shaping the representation of the theoretical framework. We have reworked this section thoroughly, as summarized in the [comment above](https://openreview.net/forum?id=3DIpIf3wQMC&noteId=ZUdGA6gH6I). We would appreciate your further feedback. Thank you!

---

### Author Response · Authors · 2022-11-18
**Simplified framework, thorough ablation study, new experiments, and clarified theoretical arguments**

Many thanks again, all reviewers, for your detailed attention and thorough, constructive feedback. Based on these, we have significantly revised our manuscript. We hope that the revised manuscript has addressed your concerns.

1. **Simplified framework, focusing exclusively on spatial attention.** To cleanly and clearly demonstrate the capability of spatial attention, we removed some of the other tricks introduced in the paper (namely Euclidean attention and augmented flow), and repeated all experiments. We hope to straightforwardly demonstrate that spatial attention is a simple yet effective framework which robustly and versatilely perform in a wide range of benchmark tasks while being significantly faster than state-of-the-art frameworks.
2. **Thorough ablation study.** Reviewers `ULzV` and `g5jj` raised concerns on the clarity of the original ablation study. We therefore conducted more thorough experiments and rewrote the ablation study section to be include in the main text, rather than the appendix. Now that the framework is simplified, we focus on the three components of SAKE—spatial attention (Eq. 6), semantic attention (Eq. 10), and speed and position update (Eq. 10), and show that spatial attention improves the model performances regardless of the rest of the choices. Reviewers `ULzV` and `g5jj`, like ourselves at the beginning, has been surprised by the underwhelming performance of EGNN on MD17. After thorough hyperparmeter search for EGNN, we added modules and tricks used in this paper, one piece after another, to EGNN, and document its performance in Appendix Table 10. The module that shows greatest impact on the EGNN performance is the distance smearing (Eq. 7) trick proposed in SchNet. Equipped with this trick the EGNN fundamentally performs like SchNet, which, too, is not able to realize angular environment.
3. **New experiments on MD17 trajectory forecast.** Reviewer `ULzV` recommended us to perform experiments on more recent and challenging tasks, namely MD17 trajectory forecast. We followed the experimental setting of GMN and reported the performance of SAKE on this task in Table 4, where SAKE outperforms all other models in 6 out of 8 systems. Note that SAKE doesn’t require hand-written ridge edges like GMN or spherical harmonics representation like TFN or SE(3)-TF.
4. **Clarified theoretical arguments.** We have rewritten the theoretical portion of the paper to clarify:
    - The universal approximation power is discussed w.r.t. the geometry space, rather than the joint space of geometry and representation (Reviewer `YAqW`)
    - Spatial attention alone maps geometry to scalars and is therefore invariant. However, these scalars can be used to parametrize equivariant functions which maps to the subspace spanned by the edge vectors. We referenced Remark 2 in the theoretical portion. (Reviewer `ULzV`) .
    - The inequality condition can rarely be violated in physical modeling as long as there are distortions as a result of vibrations in the system.

We have also added comments in the Limitations-Theoretical section of the paper to discuss the limitations of this theoretical framework—that it depends on an inequality condition and that it does not work on the node embedding space (as most GNNs do not).

---

### Decision · Program_Chairs · 2023-01-20

**Decision:**

Accept: poster

**Justification For Why Not Higher Score:**

Limited novelty, writing quality

**Justification For Why Not Lower Score:**

The method is fast and accurate, and will be of high interest to the ML for quantum chemistry community

**Metareview: Summary, Strengths And Weaknesses:**

This paper introduces an efficient equivariant network design based on spatial attention. The method performs competitively relative to various recent methods on a number of quantum chemistry benchmarks while being much faster. Since speed is one of the main selling points of ML for chemistry, this is an important result. Reviewers appreciate the simplicity and effectiveness of the approach, as well as the extensive empirical validation. Theorem 1 gives a (limited) universality result for spatial attention layers.

Several reviewers noted that the spatial attention method is a small variation of existing methods, and thus not very novel. However, given that the exact method was not known, and performs well in terms of speed and accuracy, I consider this to be a valuable contribution regardless. Another minor concern is writing quality, and I encourage the authors to continue to polish the paper.

**Note From Pc:**

if the above contains the word "oral" or "spotlight" please see: "oral" presentation means -> notable-top-5% and "spotlight" means -> notable-top-25%. As stated in our emails, we are disassociating presentation type from AC recommendations